# Mass-Rearing Conditions Do Not Always Reduce Genetic Diversity: The Case of the Mexican Fruit Fly, *Anastrepha ludens (Diptera: Tephritidae)*

**DOI:** 10.3390/insects15010056

**Published:** 2024-01-12

**Authors:** Lorena Ruiz-Montoya, Mayren Sánchez-Rosario, Emiliano López-Gómez, Maricela Garcia-Bautista, Anahí Canedo-Texón, David Haymer, Pablo Liedo

**Affiliations:** 1El Colegio de la Frontera Sur (ECOSUR), Carretera Panamericana y Periférico Sur, Barrio María Auxiliadora, San Cristóbal de las Casas 29290, Chiapas, Mexico; mgarcia@ecosur.mx; 2El Colegio de la Frontera Sur (ECOSUR), Carretera Antiguo Aeropuerto, Tapachula 30700, Chiapas, Mexico; masanchez@ecosur.edu.mx (M.S.-R.); pliedo@ecosur.mx (P.L.); 3Instituto de Biociencias, Universidad Autónoma de Chiapas, Boulevard Príncipe Akishino Sin Número Colonia Solidaridad 2000, Tapachula 30798, Chiapas, Mexico; emiliolg4@gmail.com; 4Department of Cell and Molecular Biology, University of Hawaii, 1960 East-West Rd, Biomed T511, Honolulu, HI 96822, USA; dhaymer@hawaii.edu

**Keywords:** Tephritidae, genetic differentiation, insect pest, mass-rearing adaptation, microsatellites markers, sterile insect technique

## Abstract

**Simple Summary:**

It is generally accepted that the process of insect mass-rearing, used to apply the sterile insect technique, promotes adaptation to the captive environment and a reduction in the genetic diversity of the population. Here, we compare the genetic diversity of two mass-reared strains of the Mexican fruit fly (*Anastrepha ludens*) and a wild population, using nuclear DNA. In this study, we found similar values of heterozygosity, allelic richness, and level of inbreeding among strains. These results indicate that mass-rearing conditions do not always reduce genetic diversity. Our findings contribute to understanding the genetic make-up resulting from adaptation to mass-rearing conditions.

**Abstract:**

The application of the sterile insect technique (SIT) requires the adaptation of insects to mass-rearing conditions. It is generally accepted that this adaptation may include a reduction in genetic diversity and an associated loss of desirable characteristics for the effective performance of sterile insects in the field. Here, we compare the genetic diversity of two mass-reared strains of the Mexican fruit fly, *Anastrepha ludens,* and a wild (WIL) population collected near Tapachula, Mexico, using seven DNA microsatellites as molecular genetic markers. The mass-reared strains were a bisexual laboratory strain (LAB) with approximately 130 generations under mass-rearing and a genetic sexing strain, Tapachula-7 (TA7), also under mass-rearing for 100 generations. Our results revealed an overall low level of genetic differentiation (approximately 15%) among the three strains, with the LAB and WIL populations being genetically most similar and TA7 most genetically differentiated. Although there were some differences in allele frequencies between strains, our results show that overall, the adaptation to mass-rearing conditions did not reduce genetic variability compared to the wild sample in terms of heterozygosity or allelic richness, nor did it appear to alter the level of inbreeding with respect to the wild populations. These results are contrary to the general idea that mass-rearing always results in a reduction in genetic diversity. Overall, our findings can contribute to a better understanding of the impact that adaptation to mass-rearing conditions may have on the genetic make-up of strains.

## 1. Introduction

The Mexican fruit fly, *Anastrepha ludens* (Loew) (Diptera: Tephritidae), is one of the most important agricultural pests in Mexico and Central America, particularly in fruit species of economic interest such as mango (*Mangifera indica* L.) and *Citrus* spp. fruits [1,2,3]. In 1992, the Mexican federal government established the National Campaign of Fruit Flies, and since then, the application of the sterile insect technique (SIT) has been a component of the integrated pest management strategy [4,5]. The SIT consists of the mass-rearing, sterilization, and release of millions of insects of the target pest species with the aim that sterile males will mate with wild females to induce sterility and thereby suppress the population of the pest species [6].

Two strains of *A. ludens* are mass-reared for release and control of this insect fruit pest [7]. Initial releases for the *A. ludens* SIT program consisted of both sterile males and females (from the “laboratory” strain). However, since the success of SIT depends on mating success of the sterile males, ideally, this technology will only employ the release of males as the release of sterile females is unnecessary and represents an additional cost for operational programs [8,9,10]. Accordingly, male-only releases have clearly been shown to have greater efficacy in inducing sterility in wild populations [11,12,13]. Consequently, genetic sexing strains (GSSs) have been developed where it is possible to separate the sexes at critical stages in the mass-rearing process, facilitating the release of only sterilized males [14,15].

The *A. ludens* (TA7) GSS uses alleles of a pupal color gene to distinguish males and females during the rearing process. Here, in males, the wild-type allele leading to brown pupae has been linked to the Y chromosome by a translocation. Females are homozygous for a mutant allele producing black-colored pupae, and this allows for the sexes to be sorted based on pupal color phenotype. However, as is true for genetic sexing strains from other species, the chromosomal rearrangements of the Tapachula-7 GSS are often unstable, and breakdown must be avoided to maintain the pupal color difference to separate the sexes [15,16]. For this, a “mother” colony is maintained to provide replacements via a filter rearing system, as described in [17].

Besides the use of such chromosomal rearrangements, the mass-rearing processes themselves are known to generate selection pressures (intentional and unintentional) with possible adverse effects on desirable traits, such as success in sexual competitiveness, as a byproduct of adaptation to the laboratory environment [18,19,20,21,22]. Because of this, understanding the genetic changes occurring during the mass-rearing processes can be very useful to improve colony management of strains used for SIT [23]. Hence, it is relevant to quantify the level of genetic changes associated with adaptation to a laboratory-based, constant mass-rearing environment with standardized temperature, relative humidity and a consistent light/dark photoperiod along with a high density of captive flies, to assess how much genetic diversity is lost relative to wild populations along with the degree of inbreeding promoted in captive and/or mass-reared populations.

In addition, under mass-rearing conditions such as those used for the Tapachula-7 strain (see Orozco-Davila et al. [7]), there may be greater possibilities for inbreeding and genetic drift to play important roles compared to wild populations, where selection and gene flow can be the main determining processes impacting the levels of genetic diversity [24,25]. For example, Bush et al. (1976) [26] demonstrated changes in the frequencies of two alleles of the enzyme alpha-glycerol phosphatase dehydrogenase (a-GDH) gene during colonization and mass-rearing of the screwworm *Cochliomya hominivorax*. They observed fixation (in the oldest lines), which they described as a direct result of the selection for an allelic form that works well under a mass-rearing environment. Loukas et al. (1985) [27] observed profound changes in the frequencies of two enzyme loci during the first six generations of a laboratory colony of *Bactrocera* [*Dacus*] *oleae* (Gmelin) (olive fruit fly) as a result of adaptation to laboratory conditions. Using nine microsatellite loci, Gilchrist et al. (2012) [20] found similar patterns in a strain of *Bactrocera tryoni* during the first ten generations of adaptation to mass-rearing conditions. In addition, a study by Simões et al. (2008) [28] clearly documented that genetic drift was a major force responsible for the loss of genetic variation in the adaptation of *Drosophila subobscura* collections to the laboratory environment. Many other examples have shown how colonization and adaptations to laboratory-based mass-rearing conditions have impacted genetic differentiation between domesticated and wild populations specifically for Tephritid fruit flies [20,22,27,29,30,31].

Given all of this, it is reasonable to expect that during adaptation of *A. ludens* to mass-rearing conditions for SIT application, a similar loss in genetic diversity may occur. In that context, our objective here was to compare the genetic makeup of two populations of mass-reared *A. ludens* to each other and to a wild population (WIL) without any management. For this, we used two mass-reared strains (LAB and TA7) that are maintained under different colony management strategies and a third strain of wild origin. Our expectation was that the WIL population would have greater genetic diversity than either the mass-reared LAB or TA7 strain. Our results, however, were contrary to our expectations. Here, the mass-rearing conditions appear not to have reduced the level of genetic variability compared to the wild population. We discuss the implications of this result for future management of mass-reared colonies.

## 2. Materials and Methods

### 2.1. Strains

*Anastrepha ludens* flies from three different populations were used. Flies used to establish the wild strain (WIL) were obtained as larvae from infested fruit collections of sour oranges (*Citrus aurantium*) from backyard trees scattered over an area of approximately 3 km^2^ within the surroundings of Tapachula, Chiapas, Mexico. The infested fruits were taken to the laboratory where they remained until completion of larval development. The third instar were placed in containers with vermiculite at 60% humidity to induce pupation and subsequently the emergence of adult flies.

The mass-reared bisexual laboratory strain (LAB) and the genetic sexing strain (TA7) were provided by the Planta de Cría y Esterilización de Moscas de la Fruta (Moscafrut) facility (Metapa de Domínguez, Chiapas, Mexico). Individuals from both populations were obtained at the pupal stage 48 h before adult emergence. The pupae of each strain were placed in 30 × 30 × 30 cm cages with water and a standard diet of sugar and yeast hydrolysate (MP Biomedicals, Irvine, CA, USA) in a 3:1 ratio.

The LAB strain was started approximately 13 years ago (approximately 130 generations) with individuals collected in six states of Mexico using a relatively small number of flies as founders of a new colony to replace the old colony reared at that time at the Moscafrut facility in Metapa, Chiapas. This newly established colony was refreshed only one time 9 years ago (approximately 90 generations) with approximately 4000 pupae collected in Chiapas [32]. The TA7 strain was created from a black pupal mutant seen in the LAB strain after screening approximately 7.7 million pupae. After crossing with lab-bred males, offspring were irradiated to generate chromosomal translocations linking the wild-type allele to the Y chromosome [15]. By the time of this work, the TA7 strain had approximately 100 generations under mass-rearing conditions. Approximately once a year, the TA7 strain has been subjected to further selection through the use of the “filter rearing” system [17,33] developed to eliminate translocation breakdown products that might otherwise threaten the use of this strain for genetic sexing.

For the molecular analysis, from each strain (LAB, TA7, and WIL), 30 adults (15 males and 15 females) at 15 d old were randomly collected and preserved individually in 1.5 mL vials with 95% ethanol. These were transferred to the genetics laboratory for further analysis.

### 2.2. Extraction and Amplification of Nuclear DNA

DNA extraction was performed using the 2× CTAB method [34], with some modifications. Fly heads were macerated in 1.5 mL microtubes containing 500 µL of 2× CTAB buffer. These were incubated in a water bath at 60 °C for 1 h. For DNA purification, chloroform-octanol (24:1) and isopropanol were used for precipitation. The visualization of DNA we extracted was performed using 1% agarose gel electrophoresis in 1× TAE buffer. Aliquots of 3 µL of DNA and 1.5 µL of 6× blue (loading stain) were taken and loaded into horizontal chambers. After running with a constant voltage of 90 V, the gel images were documented with a Kodak EDAS 290^®^ camera connected by an interface to a computer and stored using Id Imagen Analysis software. The analysis of diversity and genetic differentiation of *A. ludens* populations was carried out using microsatellites developed originally for *A. suspensa* [35]. Nine microsatellites (or VTR, SSR) were tested: Asus 1-H1, Asus1-2B, Asus1-2F, Asus1-3C, Asus 1-4H, Asus1-5E, Asus1-6C, Asus1-8D, and Asus1-9A (Table 1), of which only seven produced recognizable DNA fragments in *A. ludens* with the QIAxcel Advanced capillary electrophoresis system. Table 1 shows the primers used and the size ranges of products recovered at each locus.

Amplification was carried out using a T100™ Thermal Cycler BIO-RAD gradient thermocycler. PCR conditions were 94 °C for 3 min, followed by 32 cycles of 1 min at 94 °C and 30 s for primer annealing followed by 1 min at 72 °C. A final 10 min for extension at 72 °C was also added. The PCR products were visualized on 2% agarose gels. A 100 base pair (bp) molecular weight marker (ladder) was used.

The size of the amplified products was determined using the QIAXcel Advanced system for automated capillary electrophoresis, using the QIAXcel DNA High Resolution Gel Cartridge and the QIAXcel ScreenGel software version 1.6.0 (QIAGEN brand), with the QX Size Marker 25–500 bp as a reference. A total volume of 10 μL (5 μL of PCR product and 5 μL of loading buffer) was used.

Genotyping was carried out by reviewing the results obtained from the analysis of the capillary electrophoresis. This consisted of locating the peaks close to the approximate size of the product in base pairs, as reported in the literature. Positive and negative controls were used to avoid genotyping errors. For comparisons between samples, fragment sizes determined in QIAxcel were normalized and standardized using posterior analyses in the program Allelogram v. 2.2, as previously described [36].

### 2.3. Data Analyses

The genetic diversity of each sample was determined using conventional parameters including the number of alleles (Na), number of effective alleles (Ne), observed heterozygosity (Ho), unbiased expected heterozygosity (uHe), and fixation index (F) for each locus and averaged across loci. The extent of polymorphism was calculated by strain. We obtained all these parameters in GenAlEx 6.4 [37]. We tested for allelic frequency heterogeneity among strains through *G*-test heterogeneity (G) [38]. Significance differences among strains in terms of Ho and uHe were tested through ANOVA, where the Ho and uHe values were transformed by *arcsen*(H)^1/2^. Biases of average F values from zero were also tested by χ^2^ = F^2^N(k − 1), with k(k − 1) degrees of freedom [39], and we used GenAlEx 6.4 software to test for the Hardy–Weinberg equilibrium (HWE) per locus per strain.

Assessments of levels of genetic differentiation between strains were obtained from AMOVA results as well as PCA and UPGMA using GenAlEx 6.4 software [37]. The AMOVA was carried out in a hierarchical manner considering values within as well as between pairs of loci. In each case, we used 999 permutations. The θ_ST_ probability as *p* (random ≥ data) was determined based on standard permutations across the full dataset [37].

A Bayesian model was implemented in the software DIYABC v2.0 [40]. This program uses massive simulation to estimate genetic parameters. Using the DIYABC program, we tested two scenarios: (1) major differentiation between the LAB and WILD samples, and (2) between the LAB and TA7 samples. The posterior probabilities of different models were determined from the similarity between the observed dataset and the large number of simulations [41]. This allows for the estimation of genetic parameters including the mean number of alleles and the extent of differentiation as reflected in the *F*st values [40]. For the simulations, we used the simplest Generalized Stepwise Mutation model [42] for all microsatellite loci and constant effective size populations for all cases. Results were obtained using 100,000 iterations.

## 3. Results

A total of 90 individuals from each of the three strains were genetically analyzed using seven microsatellites (Appendix A). We found 46 alleles in total, and the allele frequencies per locus were not homogenous across the strains (Figure 1). The *G*-test identified significant heterogeneity for three loci (ASUS1-2F, ASUS1-4H, ASUS1-8D). Eleven private alleles were observed in the LAB strain at six loci. These comprised five private alleles at two loci in the TA7 strain along with six private alleles at five loci in the WIL strain (shown in Figure 1).

We found 100% of the loci to be polymorphic in both LAB and TA7 strains, while in the WIL strain, 85.71% were polymorphic. The highest mean number of alleles (Na) was found in the LAB (4.429 ± 0.429) strain, followed by the WIL (4.143 ± 0.829) strain and finally the TA7 (3.857 ± 0.738) strain (Figure 2A). For the number of effective alleles (Ne), the highest average value was observed in TA7 (2.23 ± 0.328), followed by the WIL (2.04 ± 0.289) and LAB (1.91 ± 0.241) strains (Figure 2A).

The average values of observed heterozygosity were 0.216, 0.271, and 0.238 for the LAB, TA7, and WIL strains, respectively. These values were 2× lower than expected for heterozygosity under EHW (Figure 2B). The ANOVA did not detect significant differences in Ho or uHe among populations. We found average F values above 0.40 for the populations, all of which were significantly different from zero (Figure 2B). All loci, with the exception of the ASUS1-6C locus in the LAB and TA7 strains, were shown to not be in Hardy–Weinberg equilibrium (Table 2).

According to the AMOVA, based on genetic Nei distances (Table 3), 85% of the genetic variation was within the strains compared to 15% among strains. The global θ_ST_ was 0.146. These results are consistent with the idea that there is a low level of genetic differentiation between the strains.

We observed low and high θ_ST_ values across loci in both overall and pairwise comparisons (Table 4). The θ_ST_ values of four loci (ASUS1-2B, ASUS1-2F, ASUS1-4H, ASUS1-8D) were significantly different from zero in the overall comparisons, indicating more substantial genetic differences, but not all loci showed significant genetic differences between all of the strains (Table 4). For example, ASUS1-1H displayed genetic differences only between LAB and WIL, and ASUS1-3B showed differences between TA7 and WIL (Table 4).

The PCA showed no obvious clustering overall. The TA7 samples tended to be biased to low values of PC1 and PC2, while the WIL and LAB samples showed negative values on PC1 but positives on PC2 (Figure 3A). The UPGMA cluster analyses indicated closer genetic similarity between the WIL and LAB strains, while the TA7 strain was more genetically distinct (Figure 3B).

Of the scenarios developed for the DIYABC analyses, our results indicate that Scenario 2, in which TA7 and LAB are the most differentiated, is the most probable, because LAB and WILD are less differentiated (Figure 4A). This conclusion is based on values of Fst for the different scenarios generated within the simulations, along with probability values, as shown in Figure 4B.

## 4. Discussion

In this study, we found generally low, but in some cases significant, genetic differentiation among two mass-reared colonies and a wild collected population (WIL) of *A. ludens* flies from Southern Mexico. One of the mass-reared strains was a standard laboratory strain (LAB) and the other was a genetic sexing strain Tapachula-7 (TA7). The TA7 strain was the most genetically differentiated, while LAB and WIL populations were genetically more similar. Overall, we found that the LAB and TA7 mass-reared strains did not show significantly reduced genetic diversity in terms of heterozygosity and allelic richness compared to the wild strain, nor did they appear to show increased levels of inbreeding compared to the wild population as measured by the parameters evaluated here.

This result contrasts with many examples of genetic differentiation between cases of wild vs. laboratory-adapted populations, characterized, in general, by loss of genetic variability upon domestication, as reported in several instances in both Tephritid and other species [29,43]. In these cases, through different mechanisms such as the founder effect, bottlenecks, inbreeding, drift, and selection pressures, colonization and artificial rearing systems are generally regarded as promoting loss of genetic diversity and genetic differentiation between mass-reared and wild strains of fruit flies [22,27,28,29,31,43,44].

Consequently, for the *A. ludens* populations considered here, we initially expected similar results. In our study, however, we found minimal genetic differences between colonized strains and wild samples. Values for some parameters such as genetic diversity (He) were slightly lower in colonized strains than wild ones, but the difference were not significant. We also found slightly more genetic similarity between the LAB and WIL strains than between the two mass-reared strains. This might reflect the common origin of the wild collection used in the establishment of the mass-reared strains, which subsequently might have been followed by the favoring of some differences found in the captive environment. Historically, the genetic sexing strain, TA7, was started from a mutant individual recovered from the LAB strain. The TA7 strain, the population undergoes repeated bottlenecks due to the use of the filter rearing system to avoid recombination, and this no doubt contributes to some genetic differentiation because these bottlenecks can be catalysts for genetic differentiation resulting from decreased genetic variation and increased inbreeding or more complex genetic interactions [45].

Consistent with all of this is the fact that our results showed that the wild samples, along with the mass-reared samples, were almost uniformly out of Hardy–Weinberg equilibrium. This was contrary to our expectation because wild populations should normally experience gene flow and other factors, processes that should not happen in the LAB and TA7 captive strains. Also, a reduction in Ho or uHe may be expected here because under laboratory conditions, artificial rearing induces intense selection pressures on a large fraction of available genetic polymorphisms. In other cases, this has led to rapid genetic differentiation between populations [27]. However, it is important note that the lack of agreement with Hardy–Weinberg expectations seen here might have resulted from the reduction in heterozygosity due to null alleles not being accounted for in our analysis [41].

Our results did show that allele frequencies of a few loci were heterogeneous among samples from all three strains, including the occurrence of significant changes in allele frequencies in at least three loci. However, this did not produce significant changes in the overall levels of genetic diversity. At least some of these frequency changes could result from processes inherent in the mass-rearing environment, including the artificial selection process, similar to changes in Adh (alcohol dehydrogenase) allele frequencies seen because of selective pressures in artificial rearing, as demonstrated in another Tephritid species, *Bactrocera oleae* [46].

We also found nearly double the number of private alleles in the LAB strain for six loci compared with TA7 or WIL. This was possibly the result of new mutations arising in populations maintained long-term (in this case, over 130 generations) even in captive conditions. Another possibility is that these private alleles could also occur at low, undetectable frequencies in the WIL strain, and that these alleles are selected for in mass-rearing conditions but not in the field [26]. Overall, other genetic diversity parameters including allele richness, effective number of alleles, and the F-fixation index were similar between the two mass-reared colonies. This may have been due, at least in part, to the common origin of both of these strains as well as the similarities in rearing environments in terms of light cycles, feeding substrates, etc.

All three samples also showed high levels of homozygosity, suggesting high levels of inbreeding. This may be unique to *A. ludens* because previous studies with wild populations reported that *A. ludens* showed high values of F-fixation [47,48,49], possibly reflecting limited dispersal of these populations and the tendency of this species to mate with related individuals, even in the field. Genetic drift may also be a relevant factor for the high levels of homozygosity since most of the samples used here were not in Hardy–Weinberg equilibrium. All three cluster analyses (PCA, UPGM, and DIYAB) also showed low levels of genetic differences among strains. Alternatively, their similarity may be understandable since the TA7 strain was derived from the LAB strain, and the LAB strain, in turn, was made up of wild collections from different regions of Mexico, including the region where the WIL strain originated from. Additionally, the diversity of wild populations may reflect founder effects if they were established by small numbers of individuals of limited genetic diversity.

## 5. Conclusions

We found that mass-rearing conditions did not result in substantially lower levels of genetic variability in terms of heterozygosity and allelic richness in comparisons of the wild and laboratory-reared strains of *A. ludens* compared here. These unexpected results are contrary to the general idea that laboratory adaptation and mass-rearing reduces overall genetic diversity compared to that seen in wild strains. It could be that many factors, including the high fecundity rate of laboratory-adapted populations, together with various selective pressures such as genetic drift, bottlenecks, and founder effects resulted in this outcome. Regardless, our findings contribute to understanding how different colony management processes may impact the genetic composition of fruit fly strains, offering useful insights that may inform colony management methods.

## Figures and Tables

**Figure 1 insects-15-00056-f001:**
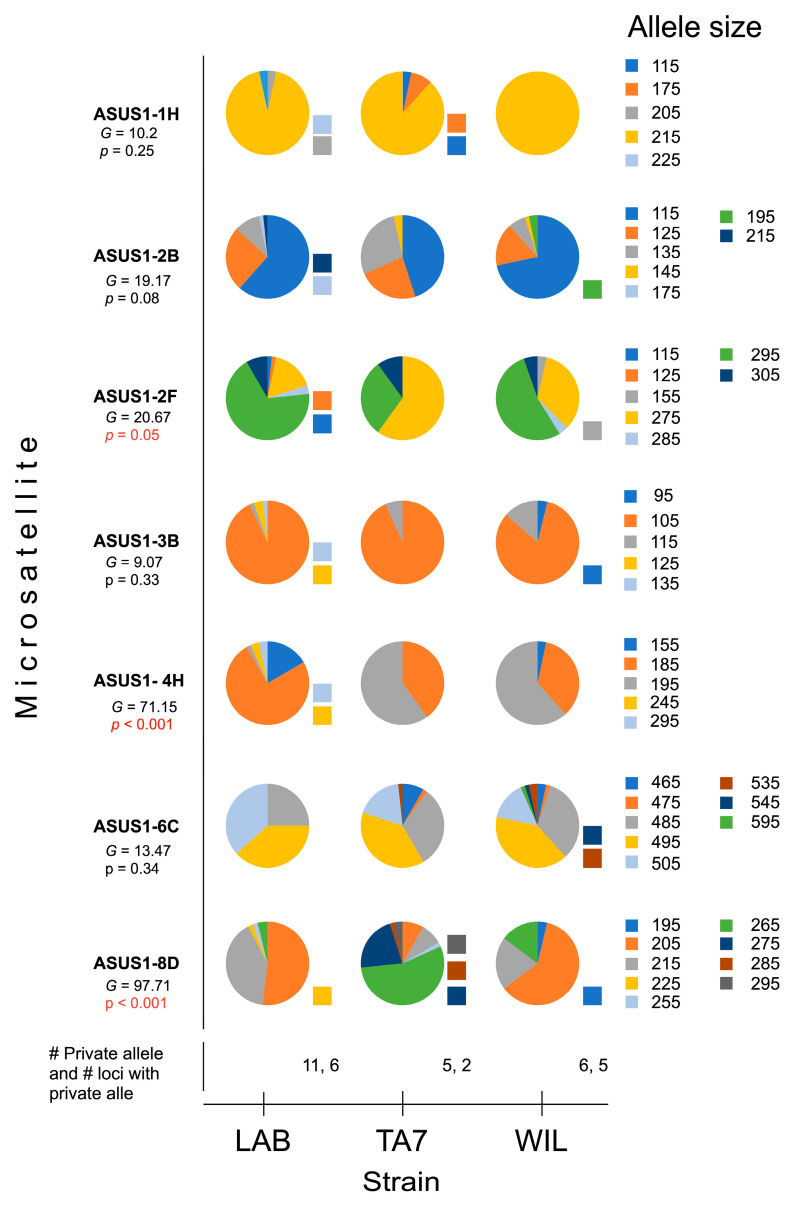
Allele frequencies of seven microsatellite loci in three strains of *Anastrepha ludens* Loew. Significant results of a G-test for heterogeneity (G) and the number of private alleles by number of loci are presented. LAB = laboratory strain; TA7 = Tapachula-7 strain; WIL = wild strain. The squares next to the pie diagrams indicate the private alleles.

**Figure 2 insects-15-00056-f002:**
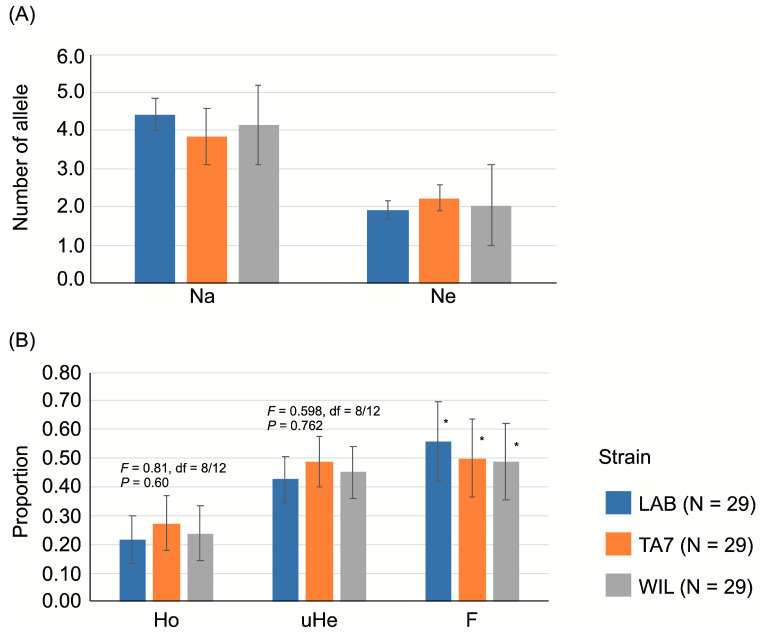
Average genetic diversity estimators for three strains of *Anastrepha ludens* Loew. (**A**) Number of alleles observed and effective. (**B**) Heterozygosity observed and unbiased expected by Hardy-Weinberg equilibrium and Fixation index. N = Average sample size, Na = observed number of alleles, Ne = effective number of alleles, uHe = unbiased expected heterozygosity, Ho = observed heterozygosity, uHe = unbiased expected heterozygosity, F = fixation index; LAB = laboratory strain, TA7 = Tapachula-7 strain, WIL = wild strain. F values from the analysis of variance (ANOVA) are presented over respective columns. An asterisk over the F bar indicates a significant result (*p* < 0.001) from the χ^2^ = F^2^N(k − 1) analysis.

**Figure 3 insects-15-00056-f003:**
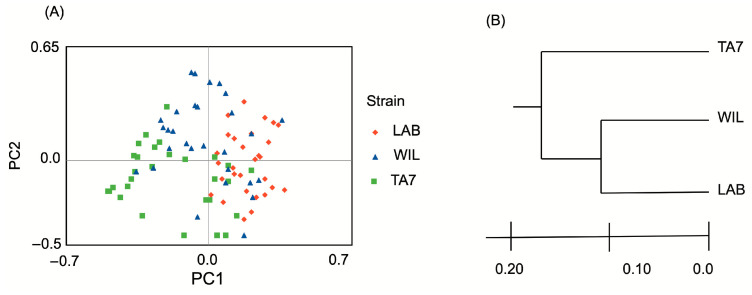
Principal component analyses (**A**) and UPGMA cluster (**B**) of genetic data of seven microsatellites in three strains of *Anastrepha ludens* Loew. LAB = laboratory strain; TA7 = Tapachula-7 strain; WIL = wild strain.

**Figure 4 insects-15-00056-f004:**
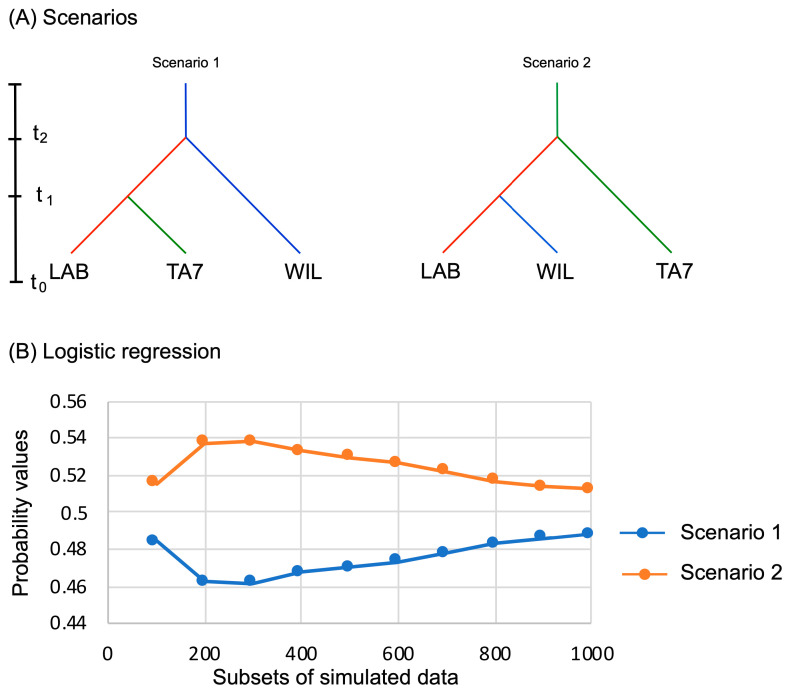
Scenarios (upper panels) of genetic differentiation of strains of *Anastrepha ludens* Loew (**A**), and probability values of logistic regression for each scenario evaluated simultaneously in DIYABC program (**B**). Time (t) is not to scale and indicates hypothetical sequences of differentiation over generations from the simulated data.

**Table 1 insects-15-00056-t001:** Microsatellites used in the study of *Anastrepha ludens* from Fritz and Schable [35].

Markers	Primer Sequence (5′-3′)	Size (bp)	Temp (°C)
Asus1-1H	1HF- TGG TAG TCA GGC ATC AA	208–230	54
	1HR- CAG TAT ATC TTG GGC AAT AA		
Asus1-2B	2BF- CAG CAG CGT ATG TAT GT	129–132	54
	2BR- CTT CGG TGT TGT TTA CTT A		
Asus1-2F	2FF- GCC ACT GGT TTA TTA CTC T	262–286	52
	2FR- ACG CCA GAC ATT TTA GTT		
Asus1-3B	3BF- CGT TCA GCA TTA CTT TGA	107–123	54
	3BR- CTT ATT TGG AAG TGA CTGA		
Asus1-4H	4HF- TGC CAT GTC TTG CTA GT	158–204	52
	4HR- TTA CCC TGA CTG ATT GTT AT		
Asus1-5E	5EF- CAA CCC GAT TCA GAT TA	235–275	52
	5ER- CGA AAA ATC CAA ATA TCT TA		
Asus1-6C	6CF- AAA TCG TGG TAA ATA AAG TAA C	333–377	54
	6CR- CGC TGC TCA ATT TAA TAC T		
Asus1-8D	8DF- GTT AAG CCA TTC CTG TTC	215–254	53
	8DR- CTG ACA GGG CAA AGT TAC		
Asus1-9A	9AF- AAA CCA TAC TTG AGA AAA AC	278–329	49
	9AR- TTG GAA CGA GAA TAA AAC		

**Table 2 insects-15-00056-t002:** Chi-squared results to test Hardy–Weinberg equilibrium in strains of *Anastrepha ludens* Loew. LAB = laboratory strain; TA7 = Tapachula strain; WIL = wild strain.

Strain	Locus	DF	ChiSq	*p*
LAB	ASUS1-1H	3	60.000	<0.01
	ASUS1-2B	10	75.820	<0.01
	ASUS1-2F	15	73.972	<0.01
	ASUS1-3B	6	56.000	<0.01
	ASUS1-4H	10	90.015	<0.01
	ASUS1-6C	3	11.422	0.010
	ASUS1-8D	10	52.095	<0.01
TA7	ASUS1-1H	3	30.267	<0.01
	ASUS1-2B	6	31.516	<0.01
	ASUS1-2F	3	21.250	<0.01
	ASUS1-3B	1	30.000	<0.01
	ASUS1-4H	1	15.648	<0.01
	ASUS1-6C	15	12.990	0.603
	ASUS1-8D	21	58.409	<0.01
WIL	ASUS1-1H	Monomorphic		
	ASUS1-2B	10	44.689	<0.01
	ASUS1-2F	10	51.854	<0.01
	ASUS1-3B	3	32.543	<0.01
	ASUS1-4H	3	41.785	<0.01
	ASUS1-6C	28	115.403	<0.01
	ASUS1-8D	6	74.851	<0.01

**Table 3 insects-15-00056-t003:** AMOVA of genetic data for the three strains of *Anastrepha ludens* Loew and the θ_ST_ estimator of genetic differentiation. The calculated θ_ST_ probability corresponds to a *p*-value (random ≥ data), based on standard permutations across the full dataset [35].

Source	df	SS	MS	Est. Var.	%	θ_ST_
Among strains	2	63.978	31.989	0.893	15%	0.146 (*p* = 0.001)
Within strains	87	453.333	5.211	5.211	85%	
Total	89	517.311		6.103	100%	

**Table 4 insects-15-00056-t004:** θ_ST_ values from the pairwise comparisons between strains of *Anastrepha ludens* Loew. The values were obtained through AMOVA per locus. LAB = laboratory strain; TA7 = Tapachula-7 strain; WIL = wild strain. The probability value (*p*) associated with θ_ST_ is based on the use of standard permutations across the full dataset.

Locus	LAB-TA7	LAB-WILL	TA7-WILL	Overall
ASUS1-1H	0.015 ns	0.017 *	0.115 ns	0.043 ns
ASUS1-2B	0.033 ns	0.0	0.105 *	0.049 *
ASUS1-1F	0.239 **	0.41 ns	0.103 **	0.158 **
ASUS1-3B	0.004 ns	0.028 ns	0.084 *	0.044 ns
ASUS1-4H	0.359 **	0.389 **	0.0	0.275 **
ASUS1-6C	0.030 *	0.043	0.0	0.015 ns
ASUS1-8D	0.297 **	0.006	0.264 *	0.049 **

ns, not significant, *p* > 0.05; *, *p* < 0.05; **, *p* < 0.01.

## Data Availability

The data presented in this study are openly available as Supplementary materials: Appendix A. Original data set of genotypes with nuclear DNA fragments of seven microsatellites of samples *Anastrapheba ludens* Loew from field and mass-reared conditions.

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
