# Peer review of "Mass-Rearing Conditions Do Not Always Reduce Genetic Diversity: The Case of the Mexican Fruit Fly, Anastrepha ludens (Diptera: Tephritidae)"

_insects, 2024, doi:10.3390/insects15010056_

Round 1
Reviewer 1 Report
Comments and Suggestions for Authors
The work by Ruiz-Montoya et al provides the fruit fly (Tephritidae) scientific community with a well-rounded explanation of their work. The study examines three populations of Anastrepha ludens - two laboratory reared and one consisting of wild fly collections. They examine the genetic diversity of each population using microsatellite loci in order to address the overall question - do mass-rearing conditions reduce the genetic diversity of SIT-reared A. ludens?
While the data, for the most part agree with their conclusion, additional information and analyses may provide clarity to some concerns.
Firstly, the authors should mention the quantity of starting material for the lab strains (lines 128-136). This is important because it may help the reader better understand the current heterozygosity/diversity. Also, it is not quite clear if and when the laboratory strains were refreshed within their history. If not refreshed by new material, please say so.
Secondly, while the authors report population conditions based on the analyses of microsatellite loci, some estimates are not reported and could reveal important outcomes. For example, in line 202, the authors mention Fst estimates but do not reveal those estimates specifically in the main body of the manuscript. They should do so. The authors in lines 327-343 are challenged with understanding the reason for wild samples being out of Hardy Weinberg equilibrium and in some cases mentioning that heterozygosity was lower than expected. One possible avenue to help understand this situation may be to estimate null alleles for those data. The presence of null alleles is one possible reason for the deficit of heterozygotes.
A few minor typos:
Line 72 - Please add "to" within "..known generate.." so that it reads ".. known to generate.."
Line 129 - Please correct "fcompatibility". Am not familiar with the term so it was probably meant to be "compatibility".
The study represents good solid science but I recommend the authors attention to these suggestions before proceeding. I thoroughly enjoyed reading the manuscript. Thank you for the opportunity to contribute to this work.
Author Response
Responses to points made by reviewer 1:
- We agree with the need to provide additional information on the origin and history of maintenance of the lab strains to better understand the current levels of heterozygosity/diversity. We have done so by adding additional information to the Materials and Methods section (lines 118-142).
- As suggested, in the results section of the manuscript (lines 292-296), we have provided additional comments on details on the Fst values scenarios generated by the DIYABC analysis and used to evaluate the different scenarios of population relationships (Figure 4)
- We also greatly appreciate the suggestions made by the reviewer to provide a better understanding of additional possible reasons for the wild samples being out of Hardy-Weinberg equilibrium. Specifically, we have added material to the manuscript (lines 339-348) discussing the potential impact of null alleles and other factors as contributing to the deficit of heterozygotes.
Additional points:
- Line 72 - text has been changed as suggested (L. 75).
- Line 129 - text has been changed.
Please see the attachment

Reviewer 2 Report
Comments and Suggestions for Authors
The document is well written, the materials and methods are clear and correctly describe the experimental work carried out, and the results are well written and agree with the data obtained. However, the document lacks the history of the LAB and Tap-7 strains. The document mentions that mass-rearing conditions are responsible for selection and adaptation, which is why heterozygosity decreases. However, it is not mentioned what they mean when they say “mass-rearing conditions.” The “founder effect” is mentioned, but it is not explained and the possible causes are not mentioned, it is not explained why the WIL and LAB strains are similar, and the possible causes of this similarity or differences are not discussed.
This conclusion “These unexpected results are contrary to the general idea that mass-rearing reduces overall genetic diversity”, is not discussed, it is only said that it is contrary to what has been obtained.
This affirmation is not supported, it does not mention what conditions, it only says this “Our findings contribute to understanding how different colony management processes may impact the genetic composition of fruit fly populations, highlighting useful knowledge for colony management methods.”
If the authors inform themselves about the mass-rearing processes, they will realize that in the Mass-rearing Facilities for fruit flies. The LAB strains continuously are enriched by introducing wild material collected in multiple locations (and the authors mention that this is the case of the strain used), which means that they not only start with high diversity, but they maintain it. On the other hand, if the authors review the works they have published on dispersal, it would seem that the flies move little, this would indicate that the genetic exchange between wild populations is very slow, and if it was only collected in 3 km2 it would seem that it is a population very small and stable. It could partially explain the results. Furthermore, the authors must consider explaining that in the case of the LAB and Tap-7 strains, although they have initially been selected by different mechanisms, these strains have reared through many generations under the same selection system; which could be giving the same results.
The topic is original and pertinent, it contributes to knowledge about the changes that occur during the colonization and mass breeding of insects and their loss of competitiveness. The techniques are well used to determine differences, although sequences that produce greater differences should have been selected during these processes. The results provide valuable information that helps to understand how to correct the selection processes.

Author Response
Responses to points made by reviewer 2:
- We agree with the need to provide additional discussion on possible reasons on how the mass rearing conditions may have contributed to the deficiency of heterozygotes seen here. We have added material to the discussion section (lines 318-324 and 339-348) to address this suggestion.
- We agree with the suggestion to provide additional support for our affirmation that our results are contrary to the general idea that mass rearing reduces overall genetic diversity, and we have added material in the Introduction section lines (74-84 and 87-93) to address this matter.
- Consistent with the suggestions made by another reviewer, we appreciate the need to provide additional information on the origin and history of the maintenance of the mass reared strains. Specifically for the mass reared strains analyzed here, we have added details to the Materials and Methods section (lines 118-126) explaining that these strains have not been continuously enriched by the introduction of wild material, and that this may be relevant to understanding the current levels of genetic diversity in these strains.
- We have also added material providing details describing that the LAB and Tap-7 strains are also not maintained under the same selection system (Materials and Methods, lines 130-142). Specifically, the maintenance of the Tap-7 strains includes the routine use of a "filter-rearing" system to maintain the integrity of the translocation linking the wild type gene for the pupal color phenotype to the Y chromosome. Without the use of this system, any products of the breakdown of the translocation would rapidly spread through the population and render useless the genetic sexing component of this strain. As we describe in this new material, the LAB strain does not undergo any similar type of routine selection.
- We agree and appreciate the suggestion that the limited rate of dispersal of the wild populations could be relevant to the lack of genetic diversity seen here because of the slow rate of genetic exchange in the wild populations and in those used to establish the mass reared strains. We have added material to the discussion (lines 367-371) to address this.
Additional points:
- Line 108,111 (and others): We changed the use of Tap-7 to TA7 to clarify the strain designations throughout the manuscript.
- Line 129 (132). Text has been changed.
Overall, we thank the reviewers for their helpful comments and efforts to provide additional clarity to the manuscript.
Please see the attachment.

Round 2
Reviewer 1 Report
Comments and Suggestions for Authors
The study provides the scientific community with well-needed and appropriately addressed hypotheses resulting in vital information for the sterile insect technique (SIT) of tephritids. I appreciate the authors taking into account those recommendations and it is my humble opinion these edits helped bring clarity and value to the manuscript.